# Physiological and Neural Changes with Rehabilitation Training in a 53-Year Amputee: A Case Study

**DOI:** 10.3390/brainsci12070832

**Published:** 2022-06-26

**Authors:** Lin Mao, Xiao Lu, Chao Yu, Kuiying Yin

**Affiliations:** 1Nanjing Research Institute of Electronic Technology, Nanjing 210019, China; mao_lin@foxmail.com (L.M.); yuchao199256@163.com (C.Y.); 2Department of Rehabilitation Medicine, The First Affiliated Hospital with Nanjing Medical University, Nanjing 210029, China; luxiao1972@163.com

**Keywords:** sEMG, EEG, rMT, amputee, prosthesis, rehabilitation

## Abstract

Many people who received amputation wear sEMG prostheses to assist in their daily lives. How these prostheses promote muscle growth and change neural activity remains elusive. We recruited a subject who had his left hand amputated for over 53 years to participate in a six-week rehabilitation training using an sEMG prosthesis. We tracked the muscle growth of his left forearm and changes in neural activity over six weeks. The subject showed an increase in fast muscle fiber in his left forearm during the training period. In an analysis of complex networks of neural activity, we observed that the α-band network decreased in efficiency but increased in its capability to integrate information. This could be due to an expansion of the network to accommodate new movements enabled by rehabilitation training. Differently, we found that in the β-band network, a band frequency related to motor functions, the efficiency of the network initially decreased but started to increase after approximately three weeks. The ability to integrate network information showed an opposite trend compared with its efficiency. rMT values, a measure that negatively correlates with cortical excitability, showed a sharp decrease in the first three weeks, suggesting an increase in cortical excitability. In the last three weeks, there was little to no change. These data indicate that rehabilitation training promoted fast muscle fiber growth and introduced neural activity changes in the subject during the first three weeks of training. Our study gave insights into how rehabilitation training with an sEMG prosthesis could lead to physiological and neural changes in amputees.

## 1. Introduction

A non-negligible number of people undergo amputation each year because of trauma, tumor, vascular disease, etc. [1]. Upper limb amputation accounts for approximately two-thirds of this population. Not only do people lose their limbs due to the amputation, but past studies reported that people who lost their limbs also showed changes in brain areas that are related to the control and sensation of their amputated hands or limbs [2]. In one study, researchers used positron emission computed tomography (PET) to monitor blood flow in the brains of amputees and controls. They found that the amputees showed a wider range of blood flow in the contralateral motor and sensory cortices when compared with control subjects [3]. More interestingly, people who received amputation for both of their upper limbs showed activation in cortical areas that are responsible for hand movements and sensation when they use their feet to perform tasks such as drawing and writing [4]. In phantom pain syndrome, a condition in which amputees experience pain coming from body parts that were no longer there, it is thought that this pain results from the functional reorganization of the brain’s sensory areas [5,6,7]. Another study showed changes in the thickness of gray matter in multiple cortical areas of the upper limb amputees [8]. Collectively, past research strongly supports the notion that the brain reorganizes in people who have an amputation.

Upper limb amputees typically choose to wear a prosthesis to aid their daily life. Currently, two types of prostheses are available: passive prostheses and active prostheses. A passive prosthesis is a cosmetic prosthesis similar to biological limbs in appearance, but it does not have additional functions. Active prostheses can be separated into two categories: (1) mechanical prostheses and (2) surface electromyography (sEMG) prostheses. The mechanical prosthesis is controlled by the movement of the shoulder that it is attached to. Usually, the “hand” part of the mechanical prosthesis is a clamp-shaped structure that appears very different from a human hand. On the contrary, an sEMG prosthesis appears similar to a human limb. An sEMG prosthesis is controlled by electromyographic signals (EMG) generated by the corresponding muscles used to control the hand. Current sEMG prostheses allow for different hand movements with multiple degrees of freedom [9,10]. In some cases where EMG cannot be collected directly from the specific muscles (e.g., people with a wider range of amputation), targeted muscle reinnervation (TMR) technology, a surgical procedure that reassigns the nerves that are used to control the limb, can be used [11]. With the development of recent technology, current research on sEMG prostheses has focused on adding somatosensory feedback to the prosthesis, complementing the motor functions [12,13].

For amputees who use a prosthesis, the prosthesis enables them to perform movements or activity that were impossible due to amputation. Their related cortices in the brain may also show changes with the introduction of the prosthesis. Indeed, numerous studies have shown changes in the brains of amputees following the use of prostheses [14,15,16]. In an fMRI study, researchers found that amputees who used prostheses daily showed more robust selective responses to images of prostheses in visual hand-selective areas of the brain [14]. Using cerebral blood perfusion as a functional indicator, Liu and her colleague (2016) found significant changes in multiple brain areas in amputees who underwent rehabilitation using prostheses [15].

Furthermore, when comparing a prosthesis with a human hand, the more similar they are, the more similar neural activity is between a control subject and an amputee using the prosthesis [16]. In this study, researchers found that amputees who used a more “natural” prosthesis that provides sensory feedback showed more similar neural activity to controls in primary sensory and motor cortices (S1 and M1) when compared with amputees who used a prosthesis without sensory feedback [16]. Taken together, these studies showed that the use of prostheses could lead to functional changes in the brains of amputees.

Most studies have focused on comparing amputees to controls or comparing amputees before and after rehabilitation, leaving it unclear when and how brain activity changes during the rehabilitation period. In the current study, we attempted to bridge this gap by continually monitoring the brain activity of an amputee who underwent a six-week rehabilitation using a prosthesis. Specifically, we recruited a subject who had his left arm amputated over 53 years ago. The subject received rehabilitation training for six weeks using an sEMG prosthesis. We measured EMG of his left forearm during the training period to monitor muscle composition changes and neural activity using EEG (electroencephalogram) and rMT (resting motor threshold). We hypothesize that there will be physiological and neural changes with the rehabilitation using a prosthesis. We also predict a time window during the training that shows the most changes in neural activity during sustained rehabilitation.

## 2. Materials and Methods

### 2.1. Participants

For this study, we recruited a subject who received a left wrist amputation due to an accident at the age of ten (age = 63, male). The subject was in healthy condition without phantom limb sensation and had not had any neurological conditions during the study. The subject showed average cognitive performance and could follow the instructions provided during the study. The subject’s residual limb is more than 90% (Figure 1), and he did not have any skin damage on the forearm. Therefore, an sEMG prosthesis developed by our lab could be used for rehabilitation training in the current study (Figure 2; also see below for details). The subject reported that he had never worn a prosthesis, never participated in similar study, and never deliberately exercised the residual limb after the amputation before. We informed him about all details regarding the training procedures prior to the study and obtained a signed consent form from him. The Ethics Committee of Jiangsu Provincial People’s Hospital approved all experimental procedures.

### 2.2. Semg Prosthesis

In this study, we used an sEMG prosthesis developed by our lab, the Linksense hand (Figure 2). Briefly, the Linksense hand collects EMG from the forearm through eight sEMG sensors placed on the targeted muscles. Then it uses algorithms to recognize the patterns of the EMG to control the prosthesis. Each finger of the hand can be controlled independently. The thumb has two degrees of freedom, whereas the rest of the fingers have one degree of freedom, adding up to six degrees of freedom. This allows for complex movements such as grabbing a complex-shaped object. Even though the functions of this prosthesis are still far from a human hand, it could still serve to improve an amputee’s quality of life significantly.

### 2.3. Prosthetic Training

In this study, the participant received rehabilitation training for six weeks. He was trained six days a week and three hours per day. Each rehabilitation training session consisted of controlling the prosthesis (using EMG) to perform multiple movements, including the following: (1) thumb bending; (2) index finger bending; (3) middle finger bending; (4) fist; (5) pinching; (6) OK gesture; (7). a combination of either the fist, pinching, or OK gesture with the thumb and index finger spread out. The subject was instructed to practice the above movements at his own pace during the training session. After the first week of training, we added more complicated movements to our training session, including movements that frequently occur in daily life (e.g., grabbing a water mug and drinking water, transporting an object, opening and closing the door). An example training session setup is shown in Figure 3.

### 2.4. Data Acquisition

#### 2.4.1. sEMG

EMG was collected by placing eight electrodes (Delsys Trigno wireless system, Natick, MA, USA) on the subject’s forearm, resting on the muscle bellies of the flexor carpi radials, extensor carpi radialis brevis, superficial digital flexor, extensor digital, extensor index finger, and flexor pollicis longus. For each experimental trial, one randomly selected movement (from the seven movements during training) was shown on the computer screen; the subject was instructed to perform the movement within 3 s using the prosthesis. Each trial was followed by a 3 s relaxation period. Every movement was repeated ten times. The subject was asked to rest for 10 s when he finished all repetitions for one movement. We repeated these procedures until all seven movements were finished (a total of 70 trials).

#### 2.4.2. EEG

We used a 64-channel EEG data acquisition device (Geodesic EEG System, EGI, Eugene, OR, USA) to monitor the EEG activity of the subject during task performance and resting. We allowed the subject to rest for approximately ten minutes before data collection to ensure that he felt comfortable. During resting-state data collection, the subject was asked to stare at a green cross in the center of the screen for two minutes with his eyes open and then closed for another two minutes. The subject was instructed to relax during the entire resting-state data collection. The resting-state session was implemented between task sessions to prevent the subject from fatigue.

#### 2.4.3. Resting Motor Threshold (rMT)

In the current study, we investigated the functions and excitability of the brain areas responsible for motor functions of the left hand using transcranial magnetic stimulation (TMS, Medtronic, Inc., Minneapolis, MN, USA). Specifically, we calculated the rMT values before, during (three weeks), and after the experiment. The rMT value refers to the minimum stimulus intensity that can induce at least 50 μV motor evoked potential (MEP) in the targeted muscle at over 50% probability [17]. It is commonly used to characterize the extent of cortical excitability changes in the short term or long term. A past study has already demonstrated that rMT can reliably represent the cortical excitability of amputees [18]. We first placed a recording electrode on the belly of the abductor digitorum brevis, a reference electrode on the styloid process of the radius, and a ground wire was placed on the lateral condyle of the humerus to record MEP. Then, we stimulated the right primary motor cortex (M1) of the subject using 50% of the maximum intensity of the TMS device. We adjusted the stimulation area in M1 until we identified a location that evoked the maximum MEP of the targeted muscles, and we then used this location as the target stimulation location. To obtain the rMT value for the targeted muscles of the subject, we tested various intensities of the TMS stimulation until we found a value that satisfied the criteria described above.

### 2.5. Data Processing

#### 2.5.1. sEMG

After amputation, patients usually experience atrophy of related muscles, leading to changes in the proportion of fast and slow muscle fibers [19,20]. The median frequency (MDF) value is a common way to detect muscle fatigue and is affected by the proportion of fast and slow muscle fibers in the targeted muscle. Past work has found that muscles that consist of a higher percentage of fast muscle fibers showed a higher value of MDF [21,22,23,24,25,26]. Consequently, an increasing value of MDF indicates an increasing proportion of fast muscle fibers and vice versa.

Fast muscle fibers are related to strength, and their discharge time is short. In comparison, slow muscle fibers are related to balance and holding static postures. Our study examined the effect of rehabilitation training on forearm muscles by measuring EMG and analyzing MDF to monitor how muscle composition changes over the course of training. Specifically, once per week during the experiment, we instructed the subject to perform a fist action ten times and recorded EMG from bilateral forearm muscles, including the flexor carpi radialis, superficial digital flexor, and thumb. The recorded EMG was filtered using a band-pass filter between 20 and 450 Hz and a notch filter at 50 Hz to remove electric hum. Then, the MDF value for each action repetition was calculated using the equation below and then averaged across repetitions:(1)$$\sum_{j=0}^{MDF}P_{j}=\sum_{j=MDF}^{M}P_{j}$$

Here, *M* refers to the size of the frequency window, and $P_{j}$ is power at frequency bin *j*.

#### 2.5.2. EEG

We first resampled EEG data to 250 Hz. Then, the data were band-pass filtered between 1 and 100 Hz, and a 50 Hz notch filter was applied to remove the electric hum. We used the averaged values of all 64 channels as our reference. Independent component analysis (ICA) was used to remove artifacts.

Complex network analysis has been widely adopted to analyze brain imaging data such as EEG, diffusion tensor imaging (DTI), functional magnetic resonance imaging (fMRI), and magnetoencephalogram (MEG) data [27,28]. We can reliably quantify functional connections of the brain networks through complex network analysis. Specifically, this analysis allows us to categorize the relationship (edges) between different electrodes or brain areas (nodes). In our study, we used different electrodes or channels in our data as different nodes and the correlation between each pair of electrodes as the value of edges between every two nodes. We first extracted α and β frequency bands from our data and performed an analysis separately. Secondly, we calculated the correlation coefficients between pairs of electrodes using a window of 2 s. Lastly, for each pair of electrodes, we averaged the correlation coefficients for each pair to obtain a single value as the edge value. The correlation coefficient was calculated as follows:(2)$$r=\frac{\sum_{i=1}^{n}\left(X_{i}-\bar{X}\right)\left(Y_{i}-\bar{Y}\right)}{\sqrt{\sum_{i=1}^{n}{\left(X_{i}-\bar{X}\right)}^{2}}\sqrt{\sum_{i=1}^{n}{\left(Y_{i}-\bar{Y}\right)}^{2}}}$$

Here, $X_{i}$ and $Y_{i}$ refer to the i-th window of data segments from the two electrodes. $\bar{X}$ and $\bar{Y}$ are the average values of the data from the two electrodes.

We employed a binary network model to investigate functional connections in our data. Specifically, we used the strongest 20% connections (high in correlation values) and assigned 1 for their connection values and 0 for the rest connections. We treated the path length between all neighboring nodes for these connections as 1. We used several network measures to characterize the change in the functional connections of the participant over the course of training.

Global efficiency is a measure to characterize a network’s capacity, information transmission and processing efficiency [29]. It is inverse to the sum of the shortest path length between all pairs of nodes and was calculated as follows: (3)$$E=\frac{1}{N}\sum_{i=1}^{N}\frac{1}{N-1}\sum_{j=1,j\neq i}^{N}\frac{1}{d_{i,j}}$$

Here, $d_{i,j}$ is the shortest path length between node *i* and node *j*, and *N* is the number of all nodes on the network.

The clustering coefficient is a measure that reflects the local density of the network [30]. For a given node i, the clustering coefficient was calculated as follows: (4)$$C_{i}=\frac{2E_{i}}{k_{i}\left(k_{i}-1\right)}$$

Here, $E_{i}$ is the number of connections that node *i* shares with other nodes, and $k_{i}$ is the number of all possible connections between node *i* and other nodes. The average clustering coefficient of all nodes in the network was calculated for this analysis: (5)$$C=\frac{1}{N}\sum_{i=1}^{N}C_{i}$$

Small-worldness is a method to characterize the ability of information integration and differentiation of the network [31] and is calculated as follows: (6)$$S=\frac{C\backslash C_{rand}}{L\backslash L_{rand}}$$
where *C* and $C_{rand}$ are the clustering coefficients of brain functional connectivity network and a random network, and *L* and $L_{rand}$ are the characteristic path lengths of brain function connection network and a random network. For random networks, we calculated the specific parameters using 1000 random networks, matching the size and connections to the complex network of our data.

## 3. Results

### 3.1. sEMG

MDF calculated from EMG is a reliable measure of different muscle fiber compositions (see the part of sEMG of Data Processing in Materials and Methods for details). We measured sEMG from the participant’s forearms before and at the end of each week’s training. Figure 4 shows the MDF values over the training period for both arms. We observed an increase in the MDF of the forearm of the left or amputated side over the training period, suggesting an increase in the proportion of fast muscle fibers. In comparison, the right or control side showed no noticeable change during the six weeks of training.

### 3.2. EEG

For out complex network analysis, we used the top 20% strongest connections of α and β bands and converted them to binary networks. Then, we computed different properties of the network across the training period as described above. These results are shown in Figure 5a,b.

For the α-band network, we observed a decreasing trend of global efficiency over the training period, suggesting that the efficiency of the α-band network decreased over the six weeks (Figure 5a). On the contrary, the small-worldness increased over time, indicating that the network’s ability to integrate and differentiate information has grown.

β-band network showed a different trend when compared with the α-band network. Specifically, we found an initial decrease in global efficiency for the first 2–3 weeks, which then increased for the rest of the experiment. Small-worldness increased for the first three weeks and then decreased for the remaining experimental period. These results suggest complex changes in the β-band network’s efficiency and ability to integrate information.

### 3.3. rMT

We computed rMT values before, during, and after rehabilitation training to assess the cortical excitability of the patient across time. The results are shown in Table 1.

During the experiment (at three weeks), we saw a sharp decrease in the rMT values from 70% to 42%, suggesting that the excitability of the motor cortex increased. However, the change for the last three weeks was minimal; the rMT value decreased from 42% to 41%, indicating that the cortical excitability reached a stable state at or before week 3.

## 4. Discussion

In this study, the subject, who had his left hand amputated for 53 years, received six weeks of rehabilitation training. We collected EMG from his forearm, EEG activity, and rMT data over the entire experiment. We hypothesized that there would be changes in his muscle composition and brain activity during the rehabilitation training.

We computed MDF based on EMG from both left and right forearms. MDF is an indicator of muscle composition, with a higher MDF indicating a higher proportion of fast muscle fibers. We observed an increase in MDF across time on the amputated side, suggesting a significant increase in fast muscle fibers with training (Figure 4). The fast muscle fibers are typically linked to strength and movement; it is possible that fast muscle fiber increased as a result of sustained hand movement training over the course of six weeks. In comparison, the MDF on the control side of the forearm showed little to no change over the six weeks (Figure 4). This could be because the subject uses his right hand regularly and has maintained a sufficient amount of fast muscle fibers.

We investigated how the subject’s brain activity changes during rehabilitation training using EEG. Specifically, we performed a complex network analysis to understand how information processing and functional connections vary with sustained training. In the α-band network, we found a decrease in the network or global efficiency and an increase in small-worldness over the training period. This means that the network showed a reduction in efficiency for information transmission but a better capacity in integrating the information (Figure 5). A possible explanation for this phenomenon is that the subject had to re-learn left-hand movements and integrate the related information or directions (e.g., muscle contractions for hand movements). The learning process led to a more complex network and thus decreased the global efficiency of the network. However, owing to a more complicated network, it can integrate and differentiate more complex information.

When examining the β-band network, we found an initial decrease in efficiency and an increase in small-worldness for the first 2–3 weeks. Then, we found an increase in efficiency and a decrease in small-worldness for the remaining time. In EEG, the β-band is related to motor and sensory events [32]. For the initial decrease in global efficiency and increase in small-worldness, it is possible that learning new movements made the network more complex and therefore decreased the efficiency but increased the capacity to integrate information (i.e., new movements). However, as the subject became more familiar with the movements, the movements could be performed easily and required less complicated information integration. Therefore, we observed an increase in efficiency and a decrease in small-worldness in the last three weeks of the experiment. Note that both network properties showed a transition at around the third week, showing a differentiated neural response for the first three weeks and the last three weeks.

In the current study, we also used TMS to obtain rMT values before, during, and after the rehabilitation training. rMT reflects the excitability of the cortex, and we specifically measured the rMT value from the contralateral motor cortex to the amputated hand. We found that the rMT sharply decreased during the first three weeks of training, indicating a strong increase in excitability of the motor cortex. This provides strong evidence that the rehabilitation training led to some degree of change or reorganization in the brain. We also found that the rMT value remained nearly unchanged for the last three weeks of training. This meshes well with our EEG data and supports our prediction of a time window for neural activity changes during rehabilitation training. This time window is at around three weeks in our data. Interestingly, in a different study with a leg amputee, the subject would sense a phantom limb for about three weeks before becoming accustomed to the new physical condition [33]. One possible explanation for these observations is that there is a time window where the brain actively undergoes functional reorganization in response to these physical changes.

In the current study, we were only able to include one subject given our restricted criteria. This subject has been amputated for an extended period (i.e., 53 years). It remains to be further investigated whether the length of time of amputation could influence the results. It is possible that the magnitude of neural and physiological changes could be influenced by time. Another possibility is that the key to the magnitude of neural and physiological changes is not time but whether the subject still has phantom limb sensation. These speculations could be evaluated in future studies.

To summarize, our data support our hypothesis that rehabilitation training with an sEMG prosthesis could lead to changes in muscle composition and neural activity. We found significant growth in fast muscle fibers that could support hand movements. In the EEG data, the α-band network showed decreased network efficiency but increased its ability to integrate information. The β-band network revealed a transition in network efficiency and capability in information integration at around the third week. Consistently, the rMT values showed a sharp decrease during the first three weeks but remained unchanged for the rest of the training period. In MDF data, our results indicate that the subject showed measurable improvements in muscle growth and changes in neural activity. In conclusion, the sEMG prosthesis training provided a satisfactory rehabilitation result in promoting muscle growth and brain reorganization in the participant. Our study provides a first look at how muscle and brain activity change over time in an amputee with sustained rehabilitation training using an sEMG prosthesis. Many open questions remain: (1) how would different training protocols influence physiological and neural changes; (2) does the length of time a subject spends as an amputee affect the changes we observe? Future studies could further investigate these questions to broaden our understanding of rehabilitation-promoted changes in amputees.

## Figures and Tables

**Figure 1 brainsci-12-00832-f001:**
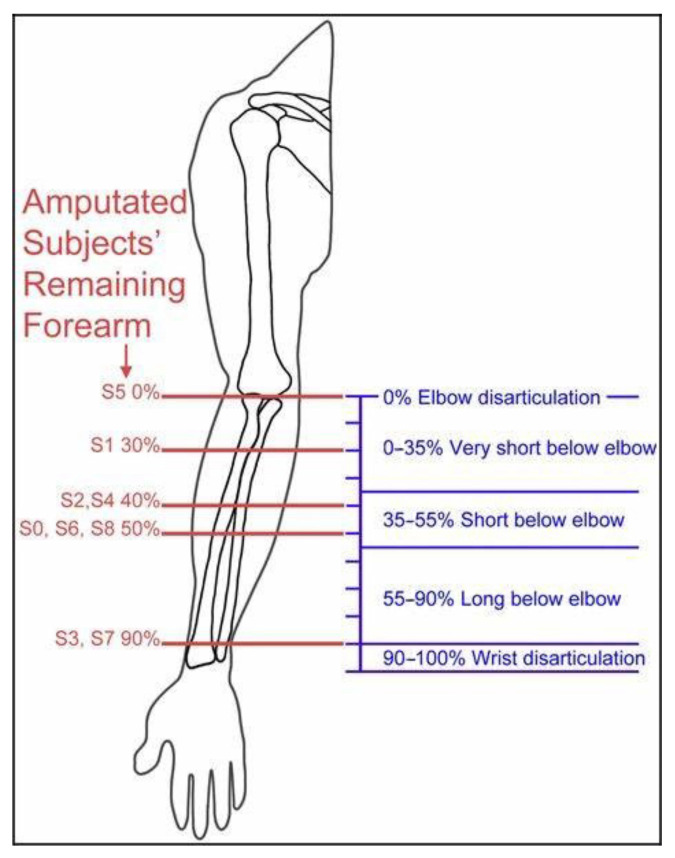
The proportion of residual limbs.

**Figure 2 brainsci-12-00832-f002:**
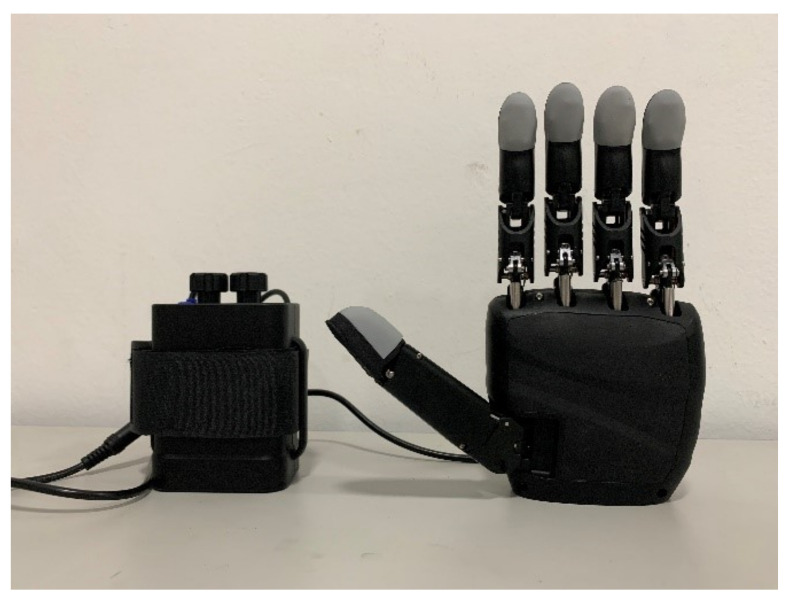
Linksense Hand.

**Figure 3 brainsci-12-00832-f003:**
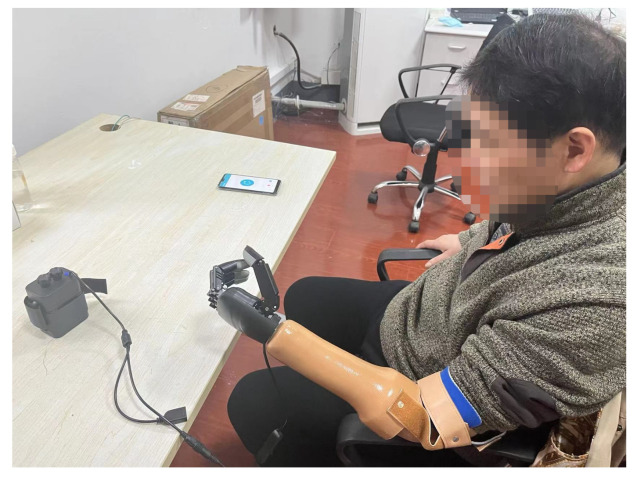
The setup of one rehabilitation training session.

**Figure 4 brainsci-12-00832-f004:**
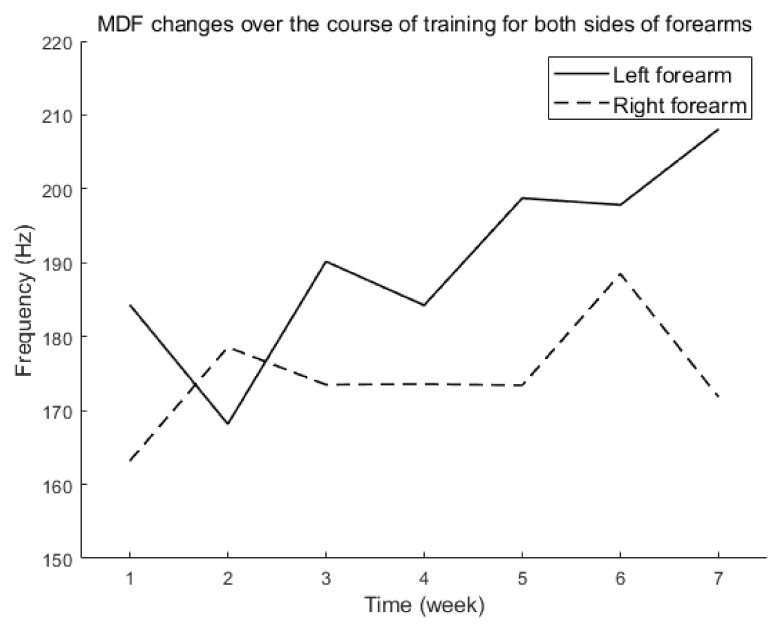
MDF changes across training period for both sides of forearms.

**Figure 5 brainsci-12-00832-f005:**
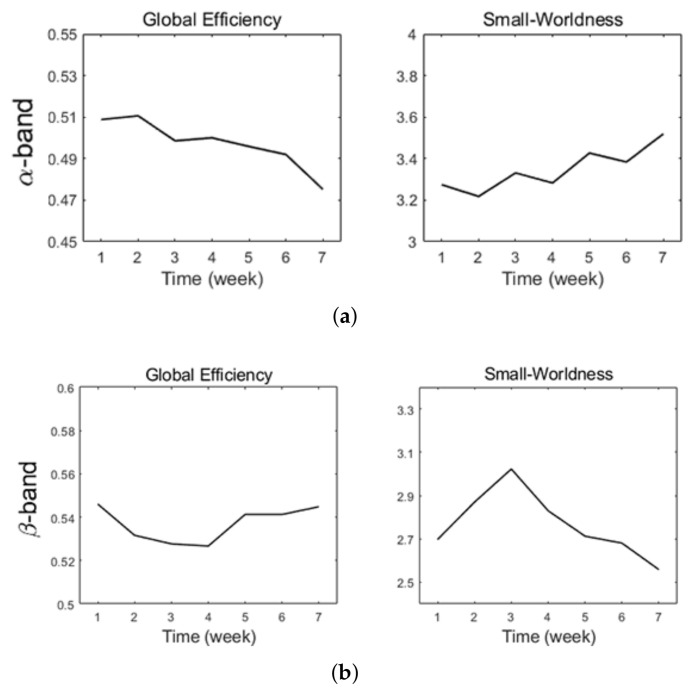
Complex network measures. (**a**) α-band network (**b**) β-band network.

**Table 1 brainsci-12-00832-t001:** The change of RMT in six weeks.

Measure Time	Stimulate Site	Collection Site	Intensity of RMT (%)
Enrollment experiment	Motor cortex M1 area	Abductor pollicis brevis	70
The third week	Motor cortex M1 area	Abductor pollicis brevis	42
The sixth week	Motor cortex M1 area	Abductor pollicis brevis	41

## Data Availability

The data are available upon reasonable request.

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
