# Peer review of "Physiological and Neural Changes with Rehabilitation Training in a 53-Year Amputee: A Case Study"

_brainsci, 2022, doi:10.3390/brainsci12070832_

Round 1

Reviewer 1 Report

The article represents a single case study in which the use of an sEMG Prosthesis was tested.
The article is well written in all its parts. Sufficient information has been given in the various sections that make it up.
However, as this is a single case study, the results obtained should be read with caution.
Some considerations should be made regarding whether the patient underwent the amputation fifty years earlier.
I believe that this work could take on a higher scientific value if it were possible to enroll at least a small group of participants, rather than a single patient with a very particular profile.

Reviewer 2 Report

The authors tracked muscle growth and changes in neural activity (measures of efficiency and capability to integrate information) associated with use of an sEMG prosthesis with 8 sEMG sensors over a 6-week training period (3 hrs/day, 6 days/week). The subject was a 63-year-old man whose left hand was amputated at age 10 (reason not specified). The authors observed an increase in fast muscle fiber in forearm, changes in alpha and beta EEG activity, and dcreased rMT (suggesting increased excitability and reorganization of motor cortex) during the first 3 weeks of training, with fewer changes thereafter.

The English grammar needs to be improved throughout the manuscript; specific cases are too numerous to list. Spelling errors (e.g., L64, “huamn hand”) need correction as well.

As the research design is case study, not an experiment, please change all (numerous) uses of the word “experiment” to “study”.

L295, I don’t think “critical period” is the appropriate term to use here. The data show that changes occur early in training—but this does mean it is a “critical period” as this term is commonly used.

The Discussion does little beyond repeating the results, repeatedly (paragraph starting on L297 summarizes the summary of results). What is needed is a discussion of how the results of this one case apply to the population of amputees using prostheses. What are the larger implications? How can the findings from this single case be applied to other contexts? And how does the fact that this subject had spent over 53 years as an amputee affect generalizability of findings? 

Author Response

请参阅附件。
